# Efficient Probabilistic Inference in the Quest for Physics Beyond the Standard Model

**Atılım Güneş Baydin,**[1] **Lukas Heinrich,**[2] **Wahid Bhimji,**[3] **Lei Shao,**[4]
**Saeid Naderiparizi,**[5] **Andreas Munk,**[5] **Jialin Liu,**[3] **Bradley Gram-Hansen,**[1] **Gilles Louppe**[6]
**Lawrence Meadows,**[4] **Philip Torr,**[1] **Victor Lee,**[4] **Prabhat,**[3] **Kyle Cranmer,**[7] **Frank Wood**[5]

[1]University of Oxford, [2]CERN, [3]Lawrence Berkeley National Lab, [4]Intel Corporation
[5]University of British Columbia, [6]University of Liege, [7]New York University

## Abstract

We present a novel probabilistic programming framework that couples directly to existing large-scale simulators through a cross-platform probabilistic execution protocol, which allows general-purpose inference engines to record and control random number draws within simulators in a language-agnostic way. The execution of existing simulators as probabilistic programs enables highly interpretable posterior inference in the structured model defined by the simulator code base. We demonstrate the technique in particle physics, on a scientifically accurate simulation of the $\tau$ (tau) lepton decay, which is a key ingredient in establishing the properties of the Higgs boson. Inference efficiency is achieved via inference compilation where a deep recurrent neural network is trained to parameterize proposal distributions and control the stochastic simulator in a sequential importance sampling scheme, at a fraction of the computational cost of a Markov chain Monte Carlo baseline.

## 1   Introduction

Complex simulators are used to express causal generative models of data across a wide segment of the scientific community, with applications as diverse as hazard analysis in seismology [49], supernova shock waves in astrophysics [36], market movements in economics [73], and blood flow in biology [72]. In these generative models, complex simulators are composed from low-level mechanistic components. These models are typically non-differentiable and lead to intractable likelihoods, which renders many traditional statistical inference algorithms irrelevant and motivates a new class of so-called likelihood-free inference algorithms [48].

There are two broad strategies for this type of likelihood-free inference problem. In the first, one uses a simulator indirectly to train a surrogate model endowed with a likelihood that can be used in traditional inference algorithms, for example approaches based on conditional density estimation [56, 70, 77, 85] and density ratio estimation [30, 35]. Alternatively, approximate Bayesian computation (ABC) [81, 87] refers to a large class of approaches for sampling from the posterior distribution of these likelihood-free models, where the original simulator is used directly as part of the inference engine. While variational inference [22] algorithms are often used when the posterior is intractable, they are not directly applicable when the likelihood of the data generating process is unknown [84].

The class of inference strategies that directly use a simulator avoids the necessity of approximating the generative model. Moreover, using a domain-specific simulator offers a natural pathway for inference algorithms to provide interpretable posterior samples. In this work, we take this approach, extend previous work in universal probabilistic programming [44, 86] and inference compilation [63, 65] to large-scale complex simulators, and demonstrate the ability to execute existing simulator codes under the control of general-purpose inference engines. This is achieved by creating a cross-

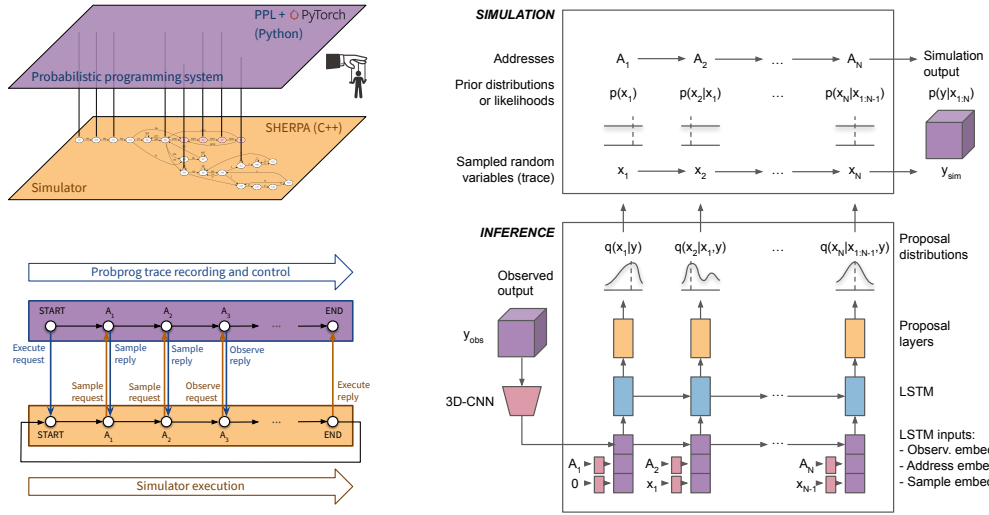

Figure 1: *Top left:* overall framework where the PPS is controlling the simulator. *Bottom left:* probabilistic execution of a single trace. *Right:* LSTM proposals conditioned on an observation.

platform probabilistic execution protocol (Figure 1, left) through which an inference engine can control simulators in a language-agnostic way. We implement a range of general-purpose inference engines from the Markov chain Monte Carlo (MCMC) [25] and importance sampling [34] families. The execution framework we develop currently has bindings in C++ and Python, which are languages of choice for many large-scale projects in science and industry. It can also be used by any other language that support flatbuffers[1] pending the implementation of a lightweight front end.

We demonstrate the technique in a particle physics setting, introducing probabilistic programming as a novel tool to determine the properties of particles at the Large Hadron Collider (LHC) [1, 29] at CERN. This is achieved by coupling our framework with SHERPA[2] [42], a state-of-the-art Monte Carlo event generator of high-energy reactions of particles, which is commonly used with Geant4[3] [5], a toolkit for the simulation of the passage of the resulting particles through detectors. In particular, we perform inference on the details of the decay of a $\tau$ (tau) lepton measured by an LHC-like detector by controlling the SHERPA simulation (with minimal modifications to the standard software), extract posterior distributions, and compare to ground truth. To our knowledge this is the first time that universal probabilistic programming has been applied in this domain and at this scale, controlling a code base of nearly one million lines of code. Our approach is scalable to more complex events and full detector simulators, paving the way to its use in the discovery of new fundamental physics.

## 2  Particle Physics and Probabilistic Inference

Our work is motivated by applications in particle physics, which studies elementary particles and their interactions using high-energy collisions created in particle accelerators such as the LHC at CERN. In this setting, collision events happen many millions of times per second, creating cascading particle decays recorded by complex detectors instrumented with millions of electronics channels. These experiments then seek to filter the vast volume of (petabyte-scale) resulting data to make discoveries that shape our understanding of fundamental physics.

The complexity of the underlying physics and of the detectors have, until now, prevented the community from employing likelihood-free inference techniques for individual collision events. However, they have developed sophisticated simulator packages such as SHERPA [42], Geant4 [5], Pythia8 [79], Herwig++ [16], and MadGraph5 [6] to model physical processes and the interactions of particles with detectors. This is interesting from a probabilistic programming point of view, because

these simulators are essentially very accurate generative models implementing the Standard Model of particle physics and the passage of particles through matter (i.e., particle detectors). These simulators are coded in Turing-complete general-purpose programming languages, and performing inference in such a setting *requires* using inference techniques developed for universal probabilistic programming that cannot be handled via more traditional inference approaches that apply to, for example, finite probabilistic graphical models [58]. Thus we focus on creating an infrastructure for the interpretation of existing simulator packages as probabilistic programs, which lays the groundwork for running inference in scientifically-accurate probabilistic models using general-purpose inference algorithms.

**The $\tau$ Lepton Decay**. The specific physics setting we focus on in this paper is the decay of a $\tau$ lepton particle inside an LHC-like detector. This is a real use case in particle physics currently under active study by LHC physicists [2] and it is also of interest due to its importance to establishing the properties of the recently discovered Higgs boson [1, 29] through its decay to $\tau$ particles [12, 33, 46, 47]. Once produced, the $\tau$ decays to further particles according to certain decay channels. The prior probabilities of these decays or "branching ratios" are shown in Figure 8 (appendix).

## 3 Related Work

### 3.1 Probabilistic Programming

Probabilistic programming languages (PPLs) extend general-purpose programming languages with constructs to do sampling and conditioning of random variables [86]. PPLs decouple model specification from inference: a model is implemented by the user as a regular program in the host programming language, specifying a model that produces samples from a generative process at each execution. In other words, the program produces samples from a joint prior distribution $p(\mathbf{x}, \mathbf{y}) = p(\mathbf{y}|\mathbf{x})p(\mathbf{x})$ that it implicitly defines, where $\mathbf{x}$ and $\mathbf{y}$ denote latent and observed random variables, respectively. The program can then be executed using a variety of general-purpose inference engines available in the PPL to obtain $p(\mathbf{x}|\mathbf{y})$, the posterior distribution of latent variables $\mathbf{x}$ conditioned on observed variables $\mathbf{y}$. *Universal* PPLs allow the expression of unrestricted probability models in a Turing-complete fashion [43, 89, 90], in contrast to languages such as Stan [28, 39] that target the more restricted model class of probabilistic graphical models [58]. Inference engines available in PPLs range from MCMC-based lightweight Metropolis Hastings (LMH) [89] and random-walk Metropolis Hastings (RMH) [62] to importance sampling (IS) [11] and sequential Monte Carlo [34]. Modern PPLs such as Pyro [20] and Edward2 [32, 82, 83] use gradient-based inference engines including variational inference [52, 57] and Hamiltonian Monte Carlo [53, 69] that benefit from modern deep learning hardware and automatic differentiation [18] features provided by PyTorch [71] and TensorFlow [3] libraries. Another way of making use of gradient-based optimization is to combine IS with deep-learning-based proposals trained with data sampled from the probabilistic program, resulting in the inference compilation (IC) algorithm [63] that enables amortized inference [40].

### 3.2 Data Analysis in Particle Physics

Inference for an individual collision event in particle physics is often referred to as reconstruction [61]. Reconstruction algorithms can be seen as a form of structured prediction: from the raw event data they produce a list of candidate particles together with their types and point-estimates for their momenta. The variance of these estimators is characterized by comparison to the ground truth values of the latent variables from simulated events. Bayesian inference on the latent state of an individual collision is rare in particle physics, given the complexity of the latent structure of the generative model. Until now, inference for the latent structure of an individual event has only been possible by accepting a drastic simplification of the high-fidelity simulators [4, 7–10, 15, 23, 27, 37, 38, 45, 59, 66, 67, 78, 80]. In contrast, inference for the fundamental parameters is based on hierarchical models and probed at the population level. Recently, machine learning techniques have been employed to learn surrogates for the implicit densities defined by the simulators as a strategy for likelihood-free inference [24].

Currently particle physics simulators are run in forward mode to produce substantial datasets that often exceed the size of datasets from actual collisions within the experiments. These are then reduced to considerably lower dimensional datasets of a handful of variables using physics domain knowledge, which can then be directly compared to collision data. Machine learning and statistical approaches for classification of particle types or regression of particle properties can be trained on these large pre-generated datasets produced by the high-fidelity simulators developed over many decades [13, 55].

The field is increasingly employing deep learning techniques allowing these algorithms to process high-dimensional, low-level data [14, 17, 31, 54, 74]. However, these approaches do not estimate the posterior of the full latent state nor provide the level of interpretability our probabilistic inference framework enables by directly tying inference results to the latent process encoded by the simulator.

# 4 Probabilistic Inference in Large-Scale Simulators

In this section we describe the main components of our probabilistic inference framework, including: (1) a novel PyTorch-based [71] PPL and associated inference engines in Python, (2) a probabilistic programming execution protocol that defines a cross-platform interface for connecting models and inference engines implemented in different languages and executed in separate processes, (3) a lighweight C++ front end allowing execution of models written in C++ under the control of our PPL.

## 4.1 Designing a PPL for Existing Large-Scale Simulators

A shortcoming of the state-of-the-art PPLs is that they are not designed to directly support *existing* code bases, requiring one to implement any model from scratch in each specific PPL. This limitation rules out their applicability to a very large body of existing models implemented as domain-specific simulators in many fields across academia and industry. A PPL, by definition, is a programming language with additional constructs for *sampling* random values from probability distributions and *conditioning* values of random variables via observations [44, 86]. Domain-specific simulators in particle physics and other fields are commonly stochastic in nature, thus they satisfy the behavior random *sampling*, albeit generally from simplistic distributions such as the continuous uniform. By interfacing with these simulators at the level of random number *sampling* (via capturing calls to the random number generator) and introducing a construct for *conditioning*, we can execute existing stochastic simulators as probabilistic programs. Our work introduces the necessary framework to do so, and makes these simulators, which commonly represent the most accurate models and understanding in their corresponding fields, subject to Bayesian inference using general-purpose inference engines. In this setting, a simulator is no longer a black box, as all predictions are directly tied into the fully-interpretable structured model implemented by the simulator code base.

To realize our framework, we implement a universal PPL called pyprob,[4] specifically designed to execute models written not only in Python but also in other languages. Our PPL currently has two families of inference engines:[5] (1) MCMC of the lightweight Metropolis–Hastings (LMH) [89] and random-walk Metropolis–Hastings (RMH) [62] varieties, and (2) sequential importance sampling (IS) [11, 34] with its regular (i.e., sampling from the prior) and inference compilation (IC) [63] varieties. The IC technique, where a recurrent neural network (NN) is trained in order to provide amortized inference to guide (control) a probabilistic program conditioning on observed inputs, forms our main inference method for performing efficient inference in large-scale simulators. Because IC training and inference uses dynamic reconfiguration of NN modules [63], we base our PPL on PyTorch [71], whose automatic differentiation feature with support for dynamic computation graphs [18] has been crucial in our implementation. The LMH and RMH engines we implement are specialized for sampling in the space of execution traces of probabilistic programs, and provide a way of sampling from the true posterior and therefore provide a baseline—at a high computational cost.

A probabilistic program can be expressed as a sequence of random samples $(x_t, a_t, i_t)_{t=1}^{T}$, where $x_t$, $a_t$, and $i_t$ are respectively the value, address,[6] and instance (counter) of a sample, the execution of which describes a joint probability distribution between latent (unobserved) random variables $\mathbf{x} := (x_t)_{t=1}^{T}$ and observed random variables $\mathbf{y} := (y_n)_{n=1}^{N}$ given by

$$p(\mathbf{x}, \mathbf{y}) := \prod_{t=1}^{T} f_{a_t}(x_t | x_{1:t-1}) \prod_{n=1}^{N} g_n(y_n | x_{\prec n}) , \tag{1}$$

where $f_{a_t}(\cdot|x_{1:t-1})$ denotes the prior probability distribution of a random variable with address $a_t$ conditional on all preceding values $x_{1:t-1}$, and $g_n(\cdot|x_{\prec n})$ is the likelihood density given the sample values $x_{\prec n}$ preceding observation $y_n$. Once a model $p(\mathbf{x}, \mathbf{y})$ is expressed as a probabilistic program, we are interested in performing inference in order to get posterior distributions $p(\mathbf{x}|\mathbf{y})$ of latent variables $\mathbf{x}$ conditioned on observed variables $\mathbf{y}$.

Inference engines of the MCMC family, designed to work in the space of probabilistic execution traces, constitute the gold standard for obtaining samples from the true posterior of a probabilistic program [62, 86, 89]. Given a current sequence of latents $\mathbf{x}$ in the trace space, these work by making proposals $\mathbf{x}'$ according to a proposal distribution $q(\mathbf{x}'|\mathbf{x})$ and deciding whether to move from $\mathbf{x}$ to $\mathbf{x}'$ based on the Metropolis–Hasting acceptance ratio of the form

$$\alpha = \min\{1, \frac{p(\mathbf{x}')q(\mathbf{x}|\mathbf{x}')}{p(\mathbf{x})q(\mathbf{x}'|\mathbf{x})}\} \; . \tag{2}$$

Inference engines in the IS family use a weighted set of samples $\{(w^k, \mathbf{x}^k)_{k=1}^K\}$ to construct an empirical approximation of the posterior distribution: $\hat{p}(\mathbf{x}|\mathbf{y}) = \sum_{k=1}^{K} w^k \delta(\mathbf{x}^k - \mathbf{x})/\sum_{j=1}^{K} w^j$, where $\delta$ is the Dirac delta function. The importance weight for each execution trace is

$$w^k = \prod_{n=1}^{N} g_n(y_n|x_{1:\tau_k(n)}^k) \prod_{t=1}^{T^k} \frac{f_{a_t}(x_t^k|x_{1:t-1}^k)}{q_{a_t, i_t}(x_t^k|x_{1:t-1}^k)} \; , \tag{3}$$

where $q_{a_t, i_t}(\cdot|x_{1:t-1}^k)$ is known as the proposal distribution and may be identical to the prior $f_{a_t}$ (as in regular IS). In the IC technique, we train a recurrent NN to receive the observed values $\mathbf{y}$ and return a set of adapted proposals $q_{a_t, i_t}(x_t|x_{1:t-1}, \mathbf{y})$ such that the approximate posterior $q(\mathbf{x}|\mathbf{y})$ is close to the true posterior $p(\mathbf{x}|\mathbf{y})$. This is achieved by using a Kullback–Leibler divergence training objective $\mathbb{E}_{p(\mathbf{y})}\left[D_{\mathrm{KL}}\left(p(\mathbf{x}|\mathbf{y}) \,||\, q(\mathbf{x}|\mathbf{y}; \phi)\right)\right]$ as

$$\mathcal{L}(\phi) := \int_{\mathbf{y}} p(\mathbf{y}) \int_{\mathbf{x}} p(\mathbf{x}|\mathbf{y}) \log \frac{p(\mathbf{x}|\mathbf{y})}{q(\mathbf{x}|\mathbf{y}; \phi)} \, \mathrm{d}\mathbf{x} \, \mathrm{d}\mathbf{y} = \mathbb{E}_{p(\mathbf{x}, \mathbf{y})}\left[-\log q(\mathbf{x}|\mathbf{y}; \phi)\right] + \mathrm{const.} \; , \tag{4}$$

where $\phi$ represents the NN weights. The weights $\phi$ are optimized to minimize this objective by continually drawing training pairs $(\mathbf{x}, \mathbf{y}) \sim p(\mathbf{x}, \mathbf{y})$ from the probabilistic program (the simulator). In IC training, we may designate a subset of all addresses $(a_t, i_t)$ to be "controlled" (learned) by the NN, leaving all remaining addresses to use the prior $f_{a_t}$ as proposal during inference. Expressed in simple terms, taking an observation $\mathbf{y}$ (an observed event that we would like to recreate or explain with the simulator) as input, the NN learns to control the random number draws of latents $\mathbf{x}$ during the simulator's execution in such a way that makes the observed outcome likely (Figure 1, right).

The NN architecture in IC is based on a stacked LSTM [51] recurrent core that gets executed for as many time steps as the probabilistic trace length. The input to this LSTM in each time step is a concatenation of embeddings of the observation $f^{\mathrm{obs}}(\mathbf{y})$, the current address $f^{\mathrm{addr}}(a_t, i_t)$, and the previously sampled value $f^{\mathrm{smp}}_{a_{t-1}, i_{t-1}}(x_{t-1})$. $f^{\mathrm{obs}}$ is a NN specific to the domain (such as a 3D convolutional NN for volumetric inputs), $f^{\mathrm{smp}}$ are feed-forward modules, and $f^{\mathrm{addr}}$ are learned address embeddings optimized via backpropagation for each $(a_t, i_t)$ pair encountered in the program execution. The addressing scheme $a_t$ is the main link between semantic locations in the probabilistic program [89] and the inputs to the NN. The address of each `sample` or `observe` statement is supplied over the execution protocol (Section 4.2) at runtime by the process hosting and executing the model. The joint proposal distribution of the NN $q(\mathbf{x}|\mathbf{y})$ is factorized into proposals in each time step $q_{a_t, i_t}$, whose type depends on the type of the prior $f_{a_t}$. In our experiments in this paper (Section 5) the simulator uses categorical and continuous uniform priors, for which IC uses, respectively, categorical and mixture of truncated Gaussian distributions as proposals parameterized by the NN. The creation of IC NNs is automatic, i.e., an open-ended number of NN modules are generated by the PPL on-the-fly when a simulator address $a_t$ is encountered for the first time during training [63]. These modules are reused (either for inference or undergoing further training) when the same address is encountered in the lifetime of the same trained NN.

A common challenge for inference in real-world scientific models, such as those in particle physics, is the presence of large dynamic ranges of prior probabilities for various outcomes. For instance, some particle decays are $\sim 10^4$ times more probable than others (Figure 8, appendix), and the prior distribution for a particle momentum can be steeply falling. Therefore some cases may be much

more likely to be seen by the NN during training relative to others. For this reason, the proposal parameters and the quality of the inference would vary significantly according to the frequency of the observations in the prior. To address this issue, we apply a "prior inflation" scheme to automatically adjust the measure of the prior distribution during training to generate more instances of these unlikely outcomes. This applies only to the training data generation for the IC NN, and the unmodified original model prior is used during inference, ensuring that the importance weights (Eq. 3) and therefore the empirical posterior are correct under the original model.

## 4.2 A Cross-Platform Probabilistic Execution Protocol

To couple our PPL and inference engines with simulators in a language-agnostic way, we introduce a probabilistic programming execution protocol (PPX)[7] that defines a schema for the execution of probabilistic programs. The protocol covers language-agnostic definitions of common probability distributions and message pairs covering the call and return values of (1) program entry points (2) `sample` statements, and (3) `observe` statements (Figure 1, left). The implementation is based on flatbuffers,[8] which is an efficient cross-platform serialization library through which we compile the protocol into the officially supported languages C++, C#, Go, Java, JavaScript, PHP, Python, and TypeScript, enabling very lightweight PPL front ends in these languages—in the sense of requiring only an implementation to call `sample` and `observe` statements over the protocol. We exchange these flatbuffers-encoded messages over ZeroMQ[9] [50] sockets, which allow seamless communication between separate processes in the same machine (using inter-process sockets) or across a network (using TCP).

Connecting any stochastic simulator in a supported language involves only the redirection of calls to the random number generator (RNG) to call the `sample` method of PPX using the corresponding probability distribution as the argument, which is facilitated when a simulator-wide RNG interface is defined in a single code file as is the case in SHERPA (Section 4.3). Conditioning is achieved by either providing an observed value for any `sample` at inference time (which means that the sample will be fixed to the observed value) or adding manual `observe` statements, similar to Pyro [20].

Besides its use with our Python PPL, the protocol defines a very flexible way of coupling any PPL system to any model so that these two sides can be (1) implemented in different programming languages and (2) executed in separate processes and on separate machines across networks. Thus we present this protocol as a probabilistic programming analogue to the Open Neural Network Exchange (ONNX)[10] project for interoperability between deep learning frameworks, in the sense that PPX is an interoperability project between PPLs allowing language-agnostic exchange of existing models (simulators). Note that, more than a serialization format, the protocol enables runtime execution of probabilistic models under the control of inference engines in different PPLs. We are releasing this protocol as a separately maintained project, together with the rest of our work in Python and C++.

## 4.3 Controlling SHERPA's Simulation of Fundamental Particle Physics

We demonstrate our framework with SHERPA [42], a Monte Carlo event generator of high-energy reactions of particles, which is a state-of-the-art simulator of the Standard Model developed by the particle physics community. SHERPA, like many other large-scale scientific projects, is implemented in C++, and therefore we implement a C++ front end for our protocol.[11] We couple SHERPA to the front end by a system-wide rerouting of the calls to the RNG, which is made easy by the existence of a third-party RNG interface (External_RNG) already present in SHERPA. Through this setup, we can repurpose, with little effort, any stochastic simulation written in SHERPA as a probabilistic generative model in which we can perform inference.

Random number draws in C++ simulators are commonly performed at a lower level than the actual prior distribution that is being simulated. This applies to SHERPA where the only samples are from the standard uniform distribution $U(0, 1)$, which subsequently get used for different purposes using transformations or rejection sampling. In our experiments (Section 5) we work with all uniform samples except for a problem-specific single address that we know to be responsible for sampling from a categorical distribution representing particle decay channels. The modification of this address

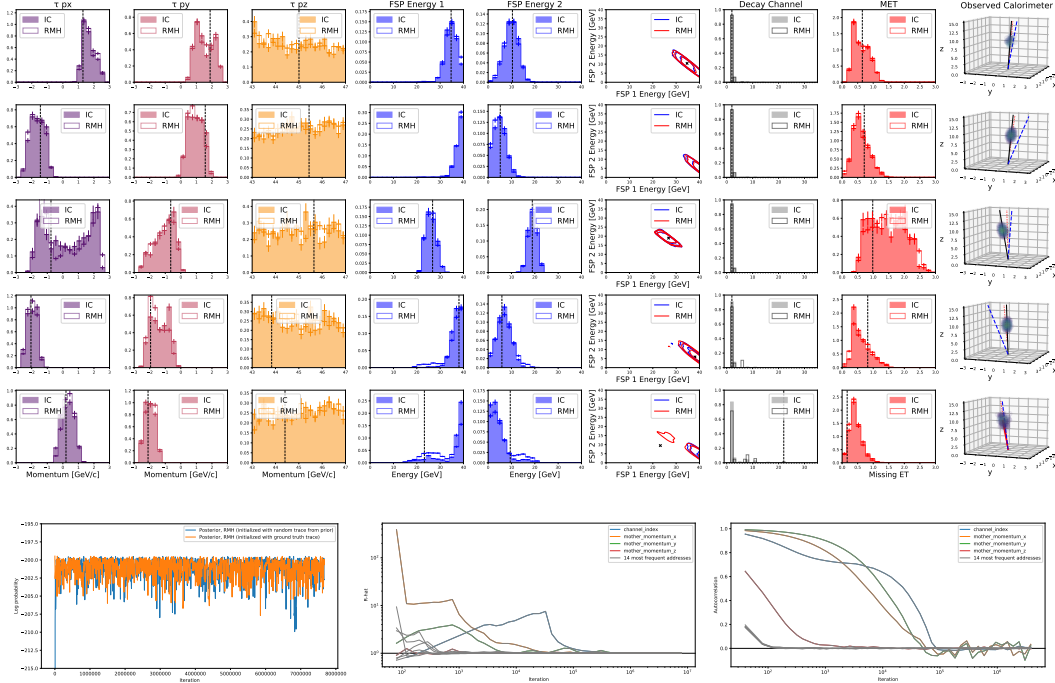

Figure 2: *Top histograms:* RMH and IC posterior results where a Channel 2 decay event ($\tau \to \nu_\tau \pi^-$) is the mode of the posterior distribution. Note that the eight variables shown are just a subset of the full latent state of several thousand addresses (Figure 5, appendix). Vertical lines indicate the point sample of the single GT trace supplying the calorimeter observation in each row. *Bottom plots:* trace joint log-probability, Gelman–Rubin diagnostic, autocorrelation results belonging to the posterior in the first row.

to use the proper categorical prior allows an effortless application of prior inflation (Section 4.1) to generate training data equally representing each channel.

Rejection sampling [41] sections in the simulator pose a challenge for our approach, as they define execution traces that are a priori unbounded; and since the IC NN has to backpropagate through every sampled value, this makes the training significantly slower. Rejection sampling is key to the application of Monte Carlo methods for evaluating matrix elements [60] and other stages of event generation in particle physics; thus an efficient treatment of this construction is primal. We address this problem by implementing a novel trace evaluation scheme which works by annotating the `sample` statements within long-running rejection sampling loops with a boolean flag called `replace`, which, when set true, enables a rejection-sampling-specific behavior for the given sample address. The simplest correct approach is to exclude these `replace` addresses from IC inference (i.e., proposing for these from the prior) and treat them as regular raw addresses in MCMC. Other approaches include amortization schemes where during IC NN training we only consider the last (thus accepted) instance $i_{\text{last}}$ of any address $(a_t, i_t)$ that fall within a rejection sampling loop. The results presented in this paper use the former simple mode. Efficient handling of rejection sampling in universal PPLs [68], and nested inference in general [75, 76], constitute an active area of research with several alternative approaches currently being formulated with varying degrees of complexity and sample efficiency that are beyond the scope of this paper.

## 5    Experiments

An important decay of the Higgs boson is to $\tau$ leptons, whose subsequent decay products interact in the detector. This constitutes a rich and realistic case to simulate, and directly connects to an important line of current research in particle physics. During simulation, SHERPA stochastically generates a set of particles to which the initial $\tau$ lepton will decay—a "decay channel"—and samples

the momenta of these particles according to a joint density obtained from underlying physical theory. These particles then interact in the detector leading to observations in the raw sensor data. While Geant4 is typically used to model the interactions in a detector, for our initial studies we implement a fast, approximate, stochastic detector simulation for a calorimeter with longitudinal and transverse segmentation (with $20{\times}35{\times}35$ voxels). The detector deposits most of the energy for electrons and $\pi^0$ into the first layers and charged hadrons (e.g., $\pi^{\pm}$) deeper into the calorimeter with larger fluctuations.

Figure 2 presents posterior distributions of a selected subset of random variables in the simulator for five different test cases where the mode of the posterior is a channel-2 decay ($\tau \rightarrow \nu_\tau \pi^-$). Test cases are generated by sampling an execution trace from the simulator prior, giving us a "ground truth trace" (GT trace), from which we extract the simulated raw 3D calorimeter as a test observation. We run our inference engines taking only these calorimeter data as input, giving us posteriors over the entire latent state of the simulator, conditioned on the observed calorimeter using a physically-motivated Poisson likelihood. We show RMH (MCMC) and IC inference results, where RMH serves as a baseline as it samples from the true posterior of the model, albeit at great computational cost. For each case, we establish the convergence of the RMH posterior to the true posterior by computing the Gelman–Rubin (GR) convergence diagnostic [26, 88] between two MCMC chains conditioned on the same observation, one starting from the GT trace and one starting from a random trace sampled from the prior.[12] As an example, in Figure 2 (bottom) we show the joint log-probability, GR diagnostic, and autocorrelation plots of the RMH posterior (with 7.7M traces) belonging to the test case in the first row. The GR result indicates that the chains converged around $10^6$ iterations, and the autocorrelation result indicates that we need approximately $10^5$ iterations to accumulate each new effectively independent sample from the true posterior. These RMH baseline results incur significant computational cost due to the sequential nature of the sampling and the large number of iterations one needs to accumulate statistically independent samples. The example we presented took 115 compute hours on an Intel E5-2695 v2 @ 2.40GHz CPU node.

We present IC posteriors conditioned on the same observations in Figure 2 and plot these together with corresponding RMH baselines, showing good agreement in all cases. These IC posteriors were obtained in less than 30 minutes in each case, representing a significant speedup compared with the RMH baseline. This is due to three main strengths of IC inference: (1) each trace executed by the IC engine gives us a statistically independent sample from the learned proposal approximating the true posterior (Equation 4) (cf. the autocorrelation time of $10^5$ in RMH); following from this independence, (2) IC inference does not necessitate a burn-in period (cf. $10^6$ iterations to convergence in GR for RMH); and (3) IC inference is embarrassingly parallelizable. These features represent the main motivation to incorporate IC in our framework to make inference in large-scale simulators computationally efficient and practicable. The results presented were obtained by running IC inference in parallel on 20 compute nodes of the type used for RMH inference, using a NN with 143,485,048 parameters that has been trained for 40 epochs with a training set of 3M traces sampled from the simulator prior, lasting two days on 32 CPU nodes. This time cost for NN training needs to be incurred only once for any given simulator setup, resulting in a trained inference NN that enables fast, repeated inference in the model specified by the simulator—a concept referred to as "amortized inference". Details of the 3DCNN–LSTM architecture used are in Figure 9 (appendix).

In the last test case in Figure 2 we show posteriors corresponding to a calorimeter observation of a Channel 22 event ($\tau \rightarrow \nu_\tau K^- K^- K^+$), a type of decay producing calorimeter depositions with similarity to Channel 2 decays and with extremely low probability in the prior (Figure 8, appendix), therefore representing a difficult case to infer. We see the posterior uncertainty in the true (RMH) posterior of this case, where Channel 2 is the mode of the posterior with a small probability mass on Channel 22 among other channels. We see that the IC posterior is able to reproduce this small probability mass on Channel 22 with success, thanks to the "prior inflation" scheme with which we train IC NNs. This leads to a proposal where Channel 22 is the mode, which later gets adjusted by importance weighting (Equation 3) to match the true posterior result (Figure 7, appendix). Our results demonstrate the feasibility of Bayesian inference in the whole latent space of this existing simulator defining a potentially unbounded number of addresses, of which we encountered approximately 24k during our experiments (Table 1 also Figure 5, appendix). To our knowledge, this is the first time a PPL system has been used with a model expressed by an existing state-of-the-art simulator at this scale.

# 6 Conclusions

We presented the first step in subsuming the vast existing body of scientific simulators, which are causal, generative models that often reflect the most accurate understanding in their respective fields, into a universal probabilistic programming framework. The ability to scale probabilistic inference to large-scale simulators is of fundamental importance to the field of probabilistic programming and the wider modeling community. It is a hard problem that requires innovations in many areas such as model–PPL interface, handling of priors with long tails, amortization of rejection sampling routines [68], addressing schemes, IC network architectures, and distributed training and inference [19] which make it difficult to cover in depth in a single paper.

Our work allows one to use existing simulator code bases to perform model-based machine learning with interpretability, where the simulator is no longer used as a black box to generate synthetic training data, but as a highly structured generative model that the simulator's code already specifies. Bayesian inference in this setting gives results that are highly interpretable, where we get to see the exact locations and processes in the model that are associated with each prediction and the uncertainty in each prediction. With this novel framework providing a clearly defined interface between domain-specific simulators and probabilistic machine learning techniques, we expect to enable a wide range of applied work straddling machine learning and fields of science and engineering. In the particle physics setting, our ultimate aim is to run the inference stage of this approach on collision data from real detectors by implementing a full LHC physics analysis together with the full posterior, so that it can be exploited for discovery of new physics via simulations that contain processes beyond the current Standard Model.

## Acknowledgments

We thank the anonymous reviewers for their constructive comments that helped us improve this paper significantly. This research used resources of the National Energy Research Scientific Computing Center (NERSC), a U.S. Department of Energy Office of Science User Facility operated under Contract No. DE-AC02-05CH11231. This work was partially supported by the NERSC Big Data Center; we acknowledge Intel for their funding support. KC, LH, and GL were supported by the National Science Foundation under the awards ACI-1450310. Additionally, KC was supported by the National Science Foundation award OAC-1836650. BGH is supported by the EPRSC Autonomous Intelligent Machines and Systems grant. AGB and PT are supported by EPSRC/MURI grant EP/N019474/1 and AGB is also supported by Lawrence Berkeley National Lab. FW is supported by DARPA D3M, under Cooperative Agreement FA8750-17-2-0093, Intel under its LBNL NERSC Big Data Center, and an NSERC Discovery grant.

## Footnotes

[1] `https://google.github.io/flatbuffers/`  [2] **S**imulation of **H**igh-**E**nergy **R**eactions of **Pa**rticles. `https://sherpa.hepforge.org/`  [3] **Ge**ometry **an**d **T**racking. `https://geant4.web.cern.ch/`

[4] `https://github.com/pyprob/pyprob`   [5] The selection of these families was motivated by working with existing simulators through an execution protocol (Section 4.2) precluding the use of gradient-based inference engines. We plan to extend this protocol in future work to incorporate differentiability.   [6] An "address" is a label uniquely identifying each sampling or conditioning event in the execution of the program. In our system it is based on a concatenation of stack frames (Table 1) leading up to the point of each random number draw, and it also includes a suffix identifying the type of associated probability distribution.

[7] `https://github.com/pyprob/ppx`    [8] `http://google.github.io/flatbuffers/`

[9] `http://zeromq.org/`    [10] `https://onnx.ai/`    [11] `https://github.com/pyprob/pyprob_cpp`

[12] The GR diagnostic compares estimated between-chains and within-chain variances, summarized as the $\hat{R}$ metric which approaches unity as the chains converge on the target distribution.

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
