[Supplementary Material]

**Efficient Probabilistic Inference in the Quest for Physics Beyond the Standard Model (Supplementary Material)**

(a) Latent probabilistic structure of the 10 most frequent trace types.

(b) Latent probabilistic structure of the 25 most frequent traces types.

(c) Latent probabilistic structure of the 100 most frequent traces types.

(d) Latent probabilistic structure of the 250 most frequent traces types.

Figure 3: Interpretability of the latent structure of the $\tau$ lepton decay process, automatically extracted from SHERPA executions via the probabilistic execution protocol. Showing model structure with increasing detail by taking an increasing number of most common trace types into account. Node labels denote address IDs (A1, A2, etc.) that correspond to uniquely identifiable parts of model execution such as those in Table 1. Addresses A1, A2, A3 correspond to momenta $p_x$, $p_y$, $p_z$, and A6 corresponds to the decay channel. Edge labels denote the frequency an edge is taken, normalized per source node. *Red:* controlled; *green*: rejection sampling; *blue*: observed; *yellow*: uncontrolled. *Note*: the addresses in these graphs are "aggregated", meaning that we collapse all instances $i_t$ of addresses $(a_t, i_t)$ into the same node in the graph, i.e., representing loops in the execution as cycles in the graph, in order to simplify the presentation. This gives us $\leq$60 aggregated addresses representing the transitions between a total of approximately 24k addresses $(a_t, i_t)$ in the simulator.

(a) Prior execution $p(\mathbf{x}, \mathbf{y})$.

(b) Posterior execution $p(\mathbf{x}|\mathbf{y})$ conditioned on a given calorimeter observation $\mathbf{y}$.

Figure 4: Interpretability of the latent probabilistic structure of the $\tau$ lepton decay simulator code, automatically extracted from 10,000 SHERPA executions via the probabilistic execution protocol. The flow is probabilistic at the shown nodes and deterministic along the edges. Edge labels denote the frequency an edge is taken, normalized per source node. *Red:* controlled; *green*: rejection sampling; *blue*: observed; *yellow*: uncontrolled. *Note*: the addresses in these graphs are "aggregated", meaning that we collapse all instances $i_t$ of addresses $(a_t, i_t)$ into the same node in the graph, i.e., representing loops in the execution as cycles in the graph, in order to simplify the presentation. This gives us $\leq 60$ aggregated addresses representing the transitions between a total of approximately 24k addresses $(a_t, i_t)$ in the simulator.

Figure 5: Example posterior over the entire latent state of the SHERPA simulator, conditioned on a single observed calorimeter. For the observation used, the posteriors presented in this figure contain approximately 6k addresses out of a total of approximately 24k addresses in the whole simulator. The histograms shown in Figure 2 are only a subset of this collection. *Note*: presenting this many plots in a single figure is challenging and a better plotting code is pending.

Figure 6: Steps of constructing the IC posterior for a Channel 2 GT event ($\tau \rightarrow \nu_\tau \pi^-$, first test case in Figure 2). The IC proposal (top row) is produced by the trained inference network. It is then weighted using Equation 3, giving IC posterior (middle row). The corresponding true posterior from RMH (MCMC) baseline is given below (bottom row). Note that the shown variables are just a subset of the full latent variables available in each case.

Figure 7: Steps of constructing the IC posterior for a Channel 22 GT event ($\tau \to \nu_\tau K^- K^- K^+$, last test case in Figure 2). The IC proposal (top row) is produced by the trained inference network. It is then weighted using Equation 3, giving IC posterior (middle row). The corresponding true posterior from RMH (MCMC) baseline is given below (bottom row). Note that the shown variables are just a subset of the full latent variables available in each case. The effect of "prior inflation" can be seen in the proposal mode of Channel 22 which the NN proposes as the most likely (i.e., mode of the proposal). However after importance weighting the IC posterior matches the true posterior from RMH (MCMC) where Channel 22 has very low (but non-zero) posterior probability due to the prior model.

Figure 8: *Top:* branching ratios of the $\tau$ lepton, effectively the prior distribution of the decay channels in SHERPA. Note that the scale is logarithmic. *Bottom:* Feynman diagrams for $\tau$ decays illustrating that these can produce multiple detected particles.

Figure 9: Training and validation losses of the IC inference NNs used for the results presented in Section 5. The network has 143,485,048 parameters and has been trained for 40 epochs. Network configuration: an LSTM with 512 hidden units; an observation embedding of size 256, encoded with a 3D convolutional NN (CNN) [64] with layer configuration Conv3D(1, 64, 3)–Conv3D(64, 64, 3)–MaxPool3D(2)–Conv3D(64, 128, 3)–Conv3D(128, 128, 3)–Conv3D(128, 128, 3)– MaxPool3D(2)–FC(2048, 256). We use previous sample embeddings of size 4 given by single-layer NNs, and address embeddings of size 64. The proposal layers are two-layer NNs, the output of which are either a mixture of ten truncated normal distributions [21] (for uniform continuous priors) or a categorical distribution (for categorical priors). We use ReLU nonlinearities in all NN components.

Table 1: Examples of addresses in the $\tau$ lepton decay problem in SHERPA (C++). Only the first 6 addresses are shown out of a total of 24,382 addresses encountered over 1,602,880 executions to collect statistics.

| Address ID | Full address |
| --- | --- |
| A1 | [forward(xt:: xarray_container<xt:: uvector<double, std:: allocator<double> >, (xt:: layout_type)1, xt:: svector<unsigned long, 4ul, std:: allocator<unsigned long>, true>, xt:: xtensor_expression_tag>)+0x5f; SherpaGenerator:: Generate()+0x36; SHERPA:: Sherpa:: GenerateOneEvent(bool)+0x2fa; SHERPA:: Event_Handler:: GenerateEvent(SHERPA:: eventtype:: code)+0x44d; SHERPA:: Event_Handler:: GenerateHadronDecayEvent(SHERPA:: eventtype:: code&)+0x45f; ATOOLS:: Random:: Get(bool, bool)+0x1d5; probprog_RNG:: Get(bool, bool)+0xf9]_Uniform_1 |
| A2 | [forward(xt:: xarray_container<xt:: uvector<double, std:: allocator<double> >, (xt:: layout_type)1, xt:: svector<unsigned long, 4ul, std:: allocator<unsigned long>, true>, xt:: xtensor_expression_tag>)+0x5f; SherpaGenerator:: Generate()+0x36; SHERPA:: Sherpa:: GenerateOneEvent(bool)+0x2fa; SHERPA:: Event_Handler:: GenerateEvent(SHERPA:: eventtype:: code)+0x44d; SHERPA:: Event_Handler:: GenerateHadronDecayEvent(SHERPA:: eventtype:: code&)+0x477; ATOOLS:: Random:: Get(bool, bool)+0x1d5; probprog_RNG:: Get(bool, bool)+0xf9]_Uniform_1 |
| A3 | [forward(xt:: xarray_container<xt:: uvector<double, std:: allocator<double> >, (xt:: layout_type)1, xt:: svector<unsigned long, 4ul, std:: allocator<unsigned long>, true>, xt:: xtensor_expression_tag>)+0x5f; SherpaGenerator:: Generate()+0x36; SHERPA:: Sherpa:: GenerateOneEvent(bool)+0x2fa; SHERPA:: Event_Handler:: GenerateEvent(SHERPA:: eventtype:: code)+0x44d; SHERPA:: Event_Handler:: GenerateHadronDecayEvent(SHERPA:: eventtype:: code&)+0x48f; ATOOLS:: Random:: Get(bool, bool)+0x1d5; probprog_RNG:: Get(bool, bool)+0xf9]_Uniform_1 |
| A4 | [forward(xt:: xarray_container<xt:: uvector<double, std:: allocator<double> >, (xt:: layout_type)1, xt:: svector<unsigned long, 4ul, std:: allocator<unsigned long>, true>, xt:: xtensor_expression_tag>)+0x5f; SherpaGenerator:: Generate()+0x36; SHERPA:: Sherpa:: GenerateOneEvent(bool)+0x2fa; SHERPA:: Event_Handler:: GenerateEvent(SHERPA:: eventtype:: code)+0x44d; SHERPA:: Event_Handler:: GenerateHadronDecayEvent(SHERPA:: eventtype:: code&)+0x8f4; ATOOLS:: Particle:: SetTime()+0xd; ATOOLS:: Flavour:: GenerateLifeTime() const+0x35; ATOOLS:: Random:: Get()+0x18b; probprog_RNG:: Get()+0xde]_Uniform_1 |
| A5 | [forward(xt:: xarray_container<xt:: uvector<double, std:: allocator<double> >, (xt:: layout_type)1, xt:: svector<unsigned long, 4ul, std:: allocator<unsigned long>, true>, xt:: xtensor_expression_tag>)+0x5f; SherpaGenerator:: Generate()+0x36; SHERPA:: Sherpa:: GenerateOneEvent(bool)+0x2fa; SHERPA:: Event_Handler:: GenerateEvent(SHERPA:: eventtype:: code)+0x44d; SHERPA:: Event_Handler:: GenerateHadronDecayEvent(SHERPA:: eventtype:: code&)+0x982; SHERPA:: Event_Handler:: IterateEventPhases(SHERPA:: eventtype:: code&, double&)+0x1d2; SHERPA:: Hadron_Decays:: Treat(ATOOLS:: Blob_List*, double&)+0x975; SHERPA:: Decay_Handler_Base:: TreatInitialBlob(ATOOLS:: Blob*, METOOLS:: Amplitude2_Tensor*, std:: vector<ATOOLS:: Particle*, std:: allocator<ATOOLS:: Particle*> > const&)+0x1ab1; SHERPA:: Hadron_Decay_Handler:: CreateDecayBlob(ATOOLS:: Particle*)+0x4cd; PHASIC:: Decay_Table:: Select() const+0x76e; ATOOLS:: Random:: Get(bool, bool)+0x1d5; probprog_RNG:: Get(bool, bool)+0xf9]_Uniform_1 |
| A6 | [forward(xt:: xarray_container<xt:: uvector<double, std:: allocator<double> >, (xt:: layout_type)1, xt:: svector<unsigned long, 4ul, std:: allocator<unsigned long>, true>, xt:: xtensor_expression_tag>)+0x5f; SherpaGenerator:: Generate()+0x36; SHERPA:: Sherpa:: GenerateOneEvent(bool)+0x2fa; SHERPA:: Event_Handler:: GenerateEvent(SHERPA:: eventtype:: code)+0x44d; SHERPA:: Event_Handler:: GenerateHadronDecayEvent(SHERPA:: eventtype:: code&)+0x982; SHERPA:: Event_Handler:: IterateEventPhases(SHERPA:: eventtype:: code&, double&)+0x1d2; SHERPA:: Hadron_Decays:: Treat(ATOOLS:: Blob_List*, double&)+0x975; SHERPA:: Decay_Handler_Base:: TreatInitialBlob(ATOOLS:: Blob*, METOOLS:: Amplitude2_Tensor*, std:: vector<ATOOLS:: Particle*, std:: allocator<ATOOLS:: Particle*> > const&)+0x1ab1; SHERPA:: Hadron_Decay_Handler:: CreateDecayBlob(ATOOLS:: Particle*)+0x4cd; PHASIC:: Decay_Table:: Select() const+0x9d7; ATOOLS:: Random:: GetCategorical(std:: vector<double, std:: allocator<double> > const&, bool, bool)+0x1a5; probprog_RNG:: GetCategorical(std:: vector<double, std:: allocator<double> > const&, bool, bool)+0x111]_Categorical(length_categories:38)_1 |