[Reviews · NeurIPS 2019]

Reviewer 1



[originality] The approaches to probabilistic inference described in the paper, such as the LMH, RMH and inference compilation IS, originate from previous work that is referred to clearly. The main contributions of this work are pulling these ideas together into a practical framework that works on a real large-scale simulator. The original challenges that are addressed include: how to apply PPL to an existing code base? how to generate diverse samples from the prior so that unlikely events can also be simulated? and how to design a protocol for the execution of probabilistic programs? [quality] It is exciting to read a paper that addresses an ambitious problem and I believe this work will inspire further research in the use of PPLs in large scale simulations in physics and elsewhere. The other strength of the paper is the sheer depth of related work that is considered and explained, while being smooth to read at the same time. Ideally, we would have had more detail on the specific contributions of this paper, particularly on the "prior inflation" scheme and the protocol. [clarity] The authors do a good job of describing how the various elements of the framework fit together. The limitations of the writing come mainly from needing further explanation and discussion for why various ideas are being used, e.g., why do you consider LMH, RMH, IC? why would you "like to employ deep neural networks" in this context? There are many methods on the shelf for probabilistic inference, why focus on these in particular? In addition, as mentioned above, more detail and space could have been given to the specific contributions of the new framework. Some further comments: + Figure 2 and the figures in the appendix are much too small to read, suggest making much larger, split across multiple figures if necessary + figures in the appendix show "interpretable graphs" but it is not clear what to interpret from these figures: how do they help a practitioner? It is hard for a non-expert to know what the value is of these graphs + "joint q(x | y)" under Eq. 3: should that read "approximate posterior" instead? + "during the simulator's execution in such a way that makes the desired outcome likely" --> suggest explaining what the "desired outcome" is + "MCMC... constitute the gold standard for obtaining samples" --> suggest explaining why it's a gold standard and whether there any limitations at all to MCMC + "via a re-routing of the random number generator" --> what do you mean by "re-routing"? + comment about variational inference not directly applicable when likelihood of data generating process is unknown --> suggest referencing the hierarchical implicit models paper (Tran et al. 2017) + a nice definition of PPL in terms of sampling and conditioning is given in Section 4.1 --> suggest moving that to where PPLs are introduced back in Section 3.1 [significance] The proposed framework can be used in future work to run inference large-scale simulations in a more-or-less plug 'n play way and the approach described can inspire other PPL methods along similar lines.

Reviewer 2



** Update ** Thanks to the authors for their elaborate response. I'm excited to see that the construction of the IC neural network is automatic, and I'm looking forward to seeing an example that illustrates how this is done, as well as other examples that give more intuition about the system. With respect to lines 18--20 of the rebuttal: I appreciate the intention to open source the system and provide URLs in the revised version of the paper. However, it is also important to provide enough details about the system in the paper itself (this also has the advantage of being a form of documentation of the system that you can link to). In addition to the points made in the review previously, I also agree with the comments made by Reviewer 3, and I hope that the revised version of the paper (for this or another conference) will include a clearer description of the PyTorch PPL, and of the work that needs to be done by the user when connecting an existing simulator to the system. ************ I am excited to see probabilistic programming applied at such a scale, and I'm looking forward to it hopefully being more widely adopted in the sciences. I didn't find any obvious mistakes in terms of technical details. My main issue with the paper, which is also why the score I give is not higher, is that it is quite difficult to read and quite abstract, leaving a lot to the reader to figure out themselves. Firstly, the paper can hugely benefit from an example of the PPL it mentions. Figure 1 provides a good intuition of how to overall framework works, but it is too abstract on its own. It is very confusing to me what the purpose of the PyTorch PPL is, as my impression was that already written simulator code is used as the probabilistic model. Other parts of the paper can also make use of some examples, e.g. the way the addressing scheme works, the rejection sampling problem, etc. Secondly, the paper gives a very good explanation of the inference compilation (IC) technique that it uses for efficient inference. However, it was not clear to me how much of it is left to the user to implement, and how much of it is automatic. For example, does the user have to build or tune the IC neural network themselves? Do they have to adjust the training procedure when "prior inflation" is needed (lines 198--208), or in the presence of rejection sampling code (lines 247--259)? If this is not automatic, what tools does the framework provide to assist with this? In general, if I have some C++ simulation code I want to use with the framework, how much extra work do I have to put in to make the cross-platform execution possible? The paper describes a probabilistic programming framework but it applies it to a single example (albeit an impressive one). Without a clear description of the system it is difficult to judge how general the framework is. ** Minor suggestions for improvement** * The elements, and especially the text of Figure 2 are too small to read. * Line 93: the PPL using TensorFlow Probability is Edward2 (Tran, 2018), which is perhaps the more accurate name and reference to use here. * Line 97: "IC" -> "inference compilation (IC)". * Reference [37]: A more up-to-date reference here is Carpenter (2017). * For papers published at a peer reviewed venue, consider citing the published paper instead of the arXiv one (e.g. [66]). * Some references are not capitalised correctly (e.g. [39]: "gibbs sampling" -> "Gibbs sampling"). Tran, Dustin, et al. "Simple, distributed, and accelerated probabilistic programming." *Advances in Neural Information Processing Systems*. 2018. Carpenter, Bob, et al. "Stan: A probabilistic programming language." *Journal of statistical software* 76.1 (2017).

Reviewer 3



Originality: The paper proposes a very novel technique to hijack a physics simulator at each sample statement and to couple it with a PPL inference engine that works closely with the original simulator to produce high quality proposals to guide inference. A number of important concepts are introduced for improving Inference Compilation (IC). These include - The concept of prior inflation to ensure that the training data for IC is diverse enough. - The concept of identifying the last accepted sample in a rejection sampling loop and using that for training the IC. - Identifying certain variables that are better off for ancestral sampling (line 180-181). This seems like a useful concept, but it would have been stronger if the paper gave some rationale or examples of these types of variables to help guide PPL researchers. - A language agnostic protocol that links any simulator code to any PPL code. Quality: There are a number of unsupported claims: The paper claims on line 51 that their work can lead to "discovery of new fundamental physics." It's unclear how this claim is justified. How do we learn new physics by studying the predictions of a simulator which has been coded up with known physics? The authors need to provide some examples or explanation supporting this claim. The paper claims that they have developed a protocol comparable to ONNX. This is a very tall claim for what is effectively an RPC with 2 or 3 function calls. The paper claims on Line 143 to have designed universal PPL inference. No further details are given about this PPL other than the fact that it supports sampling and conditioning. That level of information is not enough to understand this PPL or to ascertain if it's indeed universal. Line 238-241 is an unsupported generalization. "Differing from the conventions in the probabilistic programming community, random number draws in C++ simulators are commonly performed at a lower level than the actual prior distribution that is being simulated. This applies to SHERPA where the only samples are from the standard uniform distribution U (0, 1), which subsequently get used for different purposes using transformations or rejection sampling." For example, see reparametrization in Stan: https://mc-stan.org/docs/2_18/stan-users-guide/reparameterization-section.html Clarity: There are a number of aspects of the paper that are not very clear: - The exact interface for conditioning on data needs more explanation. Line 192 indicates that the exact address of each observe statement is supplied over the execution protocol. What is unclear is how does one map real world observations such as calorimeter readings to specific addresses in the simulator code. In a Universal PPL an observation could come from potentially different addresses depending on the execution path in the trace. The traces in the supplement show all the observations at the very end and from the same fixed addresses. This doesn't seem very general, or at the very least, there is no explanation of such a restriction in the text. - Line 190 which mentions the address embeddings doesn't seem to provide enough technical details to reproduce it. For example, word embeddings are learned from the context that they are observed in sentences either a bag of word context or a sequence context -- no such details are provided for these embeddings. - No rationale is provided for "controlled" (line 180) vs uncontrolled variables. Significance: The work is significant in providing an interface between PPL research and physics simulation as well as raising important considerations for PPL researchers in particular for the emerging area of Inference Compilation. It is possible that a researcher may simply be able to plug in a new PPL into one of these simulators and get better results without knowing anything about particle decay. This is of major importance. ** POST REBUTTAL ** I'm still not convinced that there are enough technical details in the paper. - For the "discovery of new fundamental particles" claim, I'm not a physicist so I don't understand this explanation, but I will grant the point to the authors. - The main issue remains about the lack of specification of the pytorch-based PPL for example how observations are specified how they deal with multiple control flows in which the same addresses that were observed in one trace may never be hit. Most PPLs specify some kind of abstractions to deal with this issue, but there is nothing concrete in this paper. And as I said above the control flow examples shown have all the control flows of the observations being identical. It is possible that they are some simplifications in their PPL design. This is where more details would have helped. Anyway, there is nothing in the response besides a promise to provide more details, which is not sufficient for something that goes to the crux of the paper.

[Author Response · NeurIPS 2019]

We thank the reviewers for their time and helpful remarks. Below we address some key issues and summarize the improvements we will make in the final version of the paper.

**Reviewer 1:** We thank the reviewer for their positive assessment of the paper and our work. We agree that providing more detail regarding some specific contributions would improve the paper. We will implement the following changes:

- Make better use of supplementary material space and add detailed descriptions of the "prior inflation" scheme and the protocol there, properly referenced from the main text.

- Provide further explanation for the choice of LMH, RMH, and IC inference engines and the reasons that led us focus on their use in this setting.

- Improve the text in Section 5 (Experiments) to highlight the value of the posterior results obtained in the Sherpa simulator.

- Explain how the "interpretable graphs" provided in the paper are interpreted in practice by a domain expert, by providing concrete examples of correspondence between the latent structure in the graph, the actual code implementing the model within the simulator, and the corresponding interpretation in terms of the underlying physics.

We also appreciate the detailed further suggestions about how to improve the text and figures overall, and we will implement these in the final version. We welcome the pointer to the hierarchical implicit models (Tran et al. 2017).

**Reviewer 2:** We thank the reviewer for their positive remarks and encouragement. The reviewer raises the issue that the paper's description of the framework is abstract and omits some of details. We will address the following points:

- Provide a concrete introductory example of the PPL in supplementary material. In addition to this, also note that our intention has been to make the actual Python/C++ code of the PPL and the simulator public by providing URLs in the final version of the paper. We believe that this will help the reader see and inspect the actual system in practice.

- IC inference engine is implemented in a way that is automatic, that is, the structure of the neural network is created by the system on-the-fly based on the simulator address space and the user does not have to implement a neural network for proposals. We will explain this in more detail, using the same PPL example we will introduce (previous item). We will also clearly specify the work needed by a user to connect an existing simulator for cross-platform execution.

- The PyTorch PPL serves the purpose of providing the basic PPL constructs (implementing, e.g., *sample* and *observe* statements, trace recording, inference engines) that are called from the existing simulator (implementing the probabilistic model) as a side-effect of random number sampling, through the protocol described in the paper. We will make this clearer in the text and in the figures.

We appreciate the other suggestions for improvement (text and references), which we will implement in the final version.

**Reviewer 3:** We thank the reviewer for their positive assessment together with bringing up very important points that need to be addressed to provide more technical detail in the paper. We will address the following key issues raised by the reviewer:

- The "discovery of new fundamental physics" has been the main motivation of the particle physics collaborators participating in our work. Ultimately the inference stage of the approach described here would be run on real collision data, which is largely not available for use outside LHC collaborations. Now that our approach has been demonstrated in this paper, we will work with these collaborations to implement it on a full LHC physics analysis reproducing the efficiency of point-estimates, together with the full posterior, so that this can be exploited for discovery. Note that the simulators we use also contain models of processes beyond the current Standard Model of particle physics, and the use of these parametrized simulators for building the model does not limit the ability to discover new physics when applied to real collision data in the inference stage. We will provide these explanations in the paper to support our claim about the potential future value of our work.

- The comparison between the PPL protocol and ONNX was meant as an analogy to help understand our contribution, i.e., we are introducing a project of interoperability between probabilistic programming languages by allowing a language-agnostic exchange of existing models (simulators) and PPLs. We will improve the text to clarify this.

- We will provide a more detailed introduction to our PPL using a concrete example (see Reviewer 2) and API signatures, and also improve the description of what is meant by implementing a universal PPL.

- Thank you for pointing out the reparametrization examples in Stan, we will improve our text to omit the unnecessary generalization you pointed out in line 238.

- Our PPL and protocol support conditioning on arbitrary addresses which are not restricted to be the last in the execution trace. In the Sherpa example the observations are in the end due to the conventions of using this simulator and the physics setting of conditioning on calorimeter data, which come at the end. We will provide a clear explanation of the conditioning mechanism and its implementation details.

[Meta-Review · NeurIPS 2019]

The paper presents a new probabilistic programming framework that makes Bayesian inference applicable to simulation code at scale. A large scale high energy physics application is presented. Probabilistic inference can be applied to an existing simulation code bass, allowing for ‘plug-and-play’ inference. A large-scale particle physics application was provided. On the downside, the involved inference approaches themselves have already been published before. Clarity on inference scheme could be improved; more technical details could be provided in the camera-ready version.